# Targeting Prostate Cancer, the ‘Tousled Way’

**DOI:** 10.3390/ijms241311100

**Published:** 2023-07-05

**Authors:** Siddhant Bhoir, Arrigo De Benedetti

**Affiliations:** Department of Biochemistry and Molecular Biology, LSU Health Shreveport, Shreveport, LA 71103, USA; siddhant.bhoir@lsuhs.edu

**Keywords:** PCa, AR, ADT, TLK1 signaling, and pathway inhibition

## Abstract

Androgen deprivation therapy (ADT) has been the mainstay of prostate cancer (PCa) treatment, with success in developing more effective inhibitors of androgen synthesis and antiandrogens in clinical practice. However, hormone deprivation and AR ablation have caused an increase in ADT-insensitive PCas associated with a poor prognosis. Resistance to ADT arises through various mechanisms, and most castration-resistant PCas still rely on the androgen axis, while others become truly androgen receptor (AR)-independent. Our research identified the human tousled-like kinase 1 (TLK1) as a crucial early mediator of PCa cell adaptation to ADT, promoting androgen-independent growth, inhibiting apoptosis, and facilitating cell motility and metastasis. Although explicit, the growing role of TLK1 biology in PCa has remained underrepresented and elusive. In this review, we aim to highlight the diverse functions of TLK1 in PCa, shed light on the molecular mechanisms underlying the transition from androgen-sensitive (AS) to an androgen-insensitive (AI) disease mediated by TLK1, and explore potential strategies to counteract this process. Targeting TLK1 and its associated signaling could prevent PCa progression to the incurable metastatic castration-resistant PCa (mCRPC) stage and provide a promising approach to treating PCa.

## 1. Introduction

Androgen steroid hormones play a crucial role in PCa by binding to the AR and triggering a specific oncogenic transcriptional program [1]. This understanding has been exploited for years to manage the disease’s initial or recurrent metastatic spread after surgery or radiotherapy. Despite the temporary effectiveness of hormone-deprivation therapies and anti-androgens in halting tumor growth, most patients eventually develop resistance and progress to the incurable phase of mCRPC [2,3,4,5] (Figure 1). However, targeting PCa cells early before progressing to mCRPC can significantly improve outcomes [6] (Figure 1). This can be achieved by the selective intervention of the AR-dependent and independent compensatory pathways that drive mCRPC development [7,8,9].

Recent more potent inhibitors of AR signaling (abiraterone, enzalutamide, apalutamide, and darolutamide) demonstrate some benefits for castration-resistant PCa (CRPC) patients, highlighting the critical role of AR signaling [10]. CRPC cells adapt to low androgen levels through AR gene mutations and amplifications [11], resulting in constitutively active splice variants that directly regulate the expression of DNA repair genes [12,13,14,15,16]. In addition, the upregulation of coactivators [17,18] and androgen-producing enzymes within the tumor contributes to AR signaling persistence in CRPC, making the disease drug refractory [16,19]. The complex and multifaceted nature of the mechanisms driving CRPC development and progression makes it challenging to identify effective therapies. Consequently, searching for successful CRPC treatments continues to pose a significant obstacle in PCa research.

**Figure 1 ijms-24-11100-f001:**
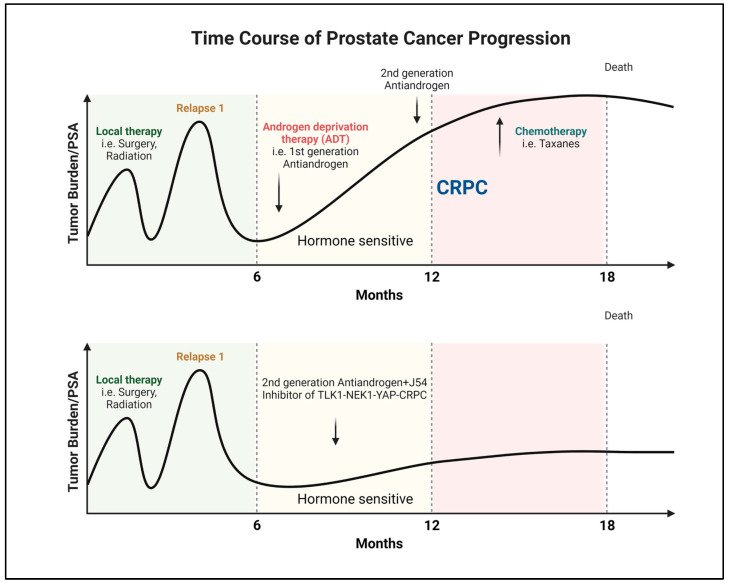
Progression and recurrence of prostate cancer measured biochemically (PSA). Created with BioRender.com.

In recent years, our laboratory has discovered that the DNA damage and response (DDR) kinase TLK1 plays a vital role in facilitating the adaptation of PCa cells to ADT (Figure 2). Initially, TLK1 promotes a cell cycle arrest by activating the TLK1-NEK1-ATR-Chk1 kinase cascade, preventing PCa cells from entering the cell cycle when faced with unfavorable growth conditions during androgen deprivation [20,21,22,23]. This arrest is a protective measure to halt replication due to the lack of androgen. Furthermore, TLK1 helps reprogram PCa cells to adapt to androgen-independent growth, which is crucial for developing CRPC. This reprogramming occurs through the NEK1-YAP/AR-CRPC conversion pathway, allowing the cells to adapt and survive with limited androgen stimulation [24,25]. TLK1 also exhibits a significant anti-apoptotic role by regulating the NEK1-VDAC1 pathway, which modulates intrinsic mitochondrial apoptotic signaling when the DDR is activated [26]. This regulation helps prevent cell death and enhances the survival of PCa cells. Additionally, our recent findings have revealed the essential role of TLK1 in promoting motility and metastasis. TLK1 indirectly regulates the kinases MK5/PRAK and AKT’s activity through AKTIP [27]. These regulatory mechanisms contribute to the increased motility and metastatic potential of PCa cells and therapy resistance.

Recognizing TLK1 as a pivotal contributor to PCa’s ability to adapt to ADT, promote growth with minimal androgen stimulation, evade apoptosis, and facilitate cell motility and metastasis offers a hopeful foundation for the ongoing quest for potential treatments. By focusing on TLK1 and its related pathways, we can develop innovative strategies to delay or halt the progression of PCa to the incurable mCRPC stage. Therefore, our endeavor to comprehend the biology of TLK1 represents a particular initiative to comprehensively understand PCa and identify appropriate therapeutic options for clinical trials.

## 2. Progression of PCa to AR-Negative Lethal Disease: Understanding the Implications

Recent advancements in inhibiting AR signaling, mainly through potent AR-targeted therapies, have resulted in a concerning trend of metastatic PCas transforming into AR-negative diseases. This change renders these tumors unresponsive to inhibition of the AR signaling pathway. While AR-negative disease is uncommon in untreated patients, its incidence has been rising in individuals receiving AR-targeted therapies [28], a trend expected to continue as these treatments become more widespread across various stages of the disease. A study conducted by Bluemn et al. revealed that the proportion of AR-negative tumors in patients with mCRPC significantly increased from 11% (1998–2011) to 36% (2012–2016) following the introduction of potent AR-targeted therapies like enzalutamide and abiraterone [28]. Although there is currently no available data, it is plausible that even the utilization of highly effective small-molecule AR degraders entering clinical trials could further elevate the percentage of AR-negative mCRPC [29,30,31].

Treatment options for AR-negative PCa, specifically the subtype with neuroendocrine differentiation, are still uncertain due to a lack of consensus. Diagnosing therapy-induced AR-negative disease is challenging as it requires assessing current tumor tissue. The recommended approach for confirmed AR-negative disease with neuroendocrine differentiation is a platinum-based regimen, similar to treatments for other neuroendocrine small cell carcinomas. However, the response rates to combinations of cisplatin and carboplatin with docetaxel or etoposide are relatively high but not long-lasting [32,33,34]. As a result, the prognosis for patients is poor, with average survival ranging from 12 to 36 months [33,35,36]. Therefore, it has become crucial to comprehend the occurrence of AR-negative PCa, which presents an urgent clinical need in the field.

## 3. Clinical Signs of AR-Negative Disease

During the progression of PCa, the loss of AR expression is accompanied by significant alterations in cellular differentiation, forming part of a more extensive cellular rewiring process [37]. This shift leads to the emergence of diverse subtypes of AR-negative PCas, each displaying distinct cellular characteristics. Identifying and distinguishing these subtypes based on morphological and molecular traits can be challenging, as some tumors may exhibit different features and gradual changes within a single nodule [38]. Consequently, the classification of AR-negative PCas remains a significant hurdle in the field.

Neuroendocrine PCa (NEPC), also known as small cell PCa, is a significant subtype among AR-negative tumors [38,39]. Neuroendocrine and basal-like proteins characterize it, while the expression of luminal and epithelial markers regulated by the AR is diminished [38]. NEPC exhibits a spectrum of histological features, ranging from well-differentiated neuroendocrine tumors to highly aggressive cancers with small cell morphology [40]. Interestingly, there are also rare histological subtypes that show squamous differentiation [41].

Newer functional analyses have demonstrated that NEPC typically originates from an AR-positive adenocarcinoma through transdifferentiation. Zou et al., in lineage-tracing mouse model studies, showed that the neuroendocrine features arise from the transdifferentiation of luminal cells [42]. Inactivating the p53 gene increased the expression of neuroendocrine markers and reduced the response to abiraterone. Using a YFP tracer under the control of the Nkx3.1 luminal-specific promoter, the study revealed that almost all tumors with neuroendocrine markers also expressed YFP, providing evidence of their luminal-epithelial origin. Additionally, androgen-sensitive prostate adenocarcinoma cells have been observed to undergo neuroendocrine differentiation in an androgen-depleted cell medium, suggesting that castration actively promotes the development of NEPC [43]. The emergence of the neuroendocrine phenotype is partly due to the suppression of AR expression, as AR increases the expression of the neuronal transcription factor BRN2 [44]. AR can directly suppress BRN2 expression, and BRN2 can modulate the activity of SOX2, a key driver of cellular plasticity [44]. Recent cancer genomics studies have also supported transdifferentiation, as NEPC often exhibits genetic alterations reminiscent of AR-dependent CRPC. These include highly recurrent AR mutations and TMPRSS2-ERG gene fusions [36,45,46,47]. It is worth noting that these TMPRSS2-ERG gene rearrangements activate the transcriptional program regulated by YAP1 and that prostate-specific activation of either ERG or YAP1 in mice induces similar transcriptional changes and results in age-related prostate tumors [24,48,49]. These findings further emphasize the significance of targeting the TLK1-NEK1-YAP pathway during early adaptation to ADT.

Recent findings have revealed a connection between the activation of Wnt/β-Catenin signaling and the presence of AR-negative disease. This connection is established through the involvement of specific genes regulated by Wnt/β-Catenin, namely FOXA2, and MYCN, which promote the process of neuroendocrine transdifferentiation [50,51,52,53,54]. Laboratory experiments conducted on PCa cells have indicated that the presence of active β-Catenin leads to an increase in the production of neuroendocrine-specific proteins like NSE and chromogranin A [55]. Interestingly, inhibiting Wnt/β-Catenin signaling has decreased neuroendocrine transdifferentiation in laboratory settings [55]. Additionally, the upregulation of Wnt-11 has been observed in NEPC, and it has been functionally linked to the transdifferentiation process in vitro [56].

A recently identified subtype of AR-negative PCa, the double-negative subtype, has become more prevalent in recent years. This subtype, which lacks AR and neuroendocrine markers, has shown a strong association with the use of newer AR pathway inhibitors [28]. Its occurrence has increased from 5% to over 20% in the past decade, making it one of the most common types of AR-negative PCa [28]. The growth of these double-negative tumors is supported by heightened autocrine FGF signaling, which activates the MAPK pathway and contributes to their proliferation [28]. Consequently, these cancer cells exhibit sensitivity to pharmacological inhibition of FGF and MAPK signaling.

A newly discovered subtype of mCRPC has been linked to the loss of chromodomain helicase DNA-binding protein 1 (CHD1). When CHD1 is deficient, cancer cells undergo significant changes in their chromatin structure, making them resistant to enzalutamide [57]. These resistant tumors often exhibit higher glucocorticoid receptor (GR) expression, leading to sustained signaling that promotes drug resistance. However, inhibiting GR in the presence of CHD1 deficiency can restore sensitivity to enzalutamide [57], indicating that increased GR activity is crucial in developing resistance to this drug.

Labrecque et al. recently identified five mCRPC subtypes based on AR and neuroendocrine marker RNA expression patterns. These subtypes include tumors with high AR expression (ARPC), low AR expression (ARLPC), tumors expressing both AR and neuroendocrine markers (AMPC), tumors lacking both AR and neuroendocrine markers (DNPC), and tumors exhibiting small cell and neuroendocrine features without AR expression (SCNPC) [58]. Although progress has been made, further research is needed to understand the similarities and differences among these AR-negative subtypes regarding clinical characteristics and potential treatment options. This will require collaborative efforts from multidisciplinary teams, the integration of molecular biomarkers, and precise patient selection, as highlighted in a recent workshop sponsored by the National Cancer Institute (NCI) [59].

The subtypes mentioned earlier are only sometimes distinct from each other and can change over time [58]. They likely represent stages of a continuous process where cancer cells lose their specialized characteristics, acquire stem cell-like properties, and eventually become independent of and resistant to AR signaling [58,60,61].

## 4. Human TLKs, Their Substrate Interaction, and Functional Significance

Tousled-like kinases (TLKs) are mammalian homologs of plant-tousled kinases, first discovered in 1999 [62]. They play a crucial role in DNA replication, particularly during the S-phase of the cell cycle. The TLK family consists of two members: TLK1 and TLK2. These kinases have similar substrates and partially overlap in maintaining genomic stability. TLK1, in particular, has been extensively studied and found to be involved in various cellular processes, including DNA replication, transcription, cell cycle regulation, DNA damage response, and repair [63,64,65].

One specific isoform of TLK1 called TLK1B has been very well characterized. TLK1B is a spliced variant with a truncated N-terminal region, a long GC-rich 5’ untranslated region (UTR), and two short upstream open reading frames (ORFs) inhibitory for translation. However, TLK1B retains the intact catalytic domain of TLK1 and exhibits similar substrate specificity [64,66,67]. Therefore, throughout this review, we have used the terms “TLK1” and “TLK1B” interchangeably. The translation of TLK1B is facilitated by the overexpression of eIF4E or activation of the AKT-mTOR pathway, which leads to the release of eIF4E from its inhibitor [66,68,69].

TLK1B plays a role in phosphorylating histone H3 at the Serine-10 residue [70]. This phosphorylation event promotes proper chromosomal condensation during metaphase and provides radioresistance in healthy mouse breast epithelial cells (MM3MG) and yeast [70]. TLK1B also phosphorylates the histone H3-H4 chaperones Asf1a and Asf1b, enhancing their affinity to bind to H3-H4 and enabling nucleosomal assembly during replication and damage repair [71,72]. Moreover, TLK1B phosphorylates Rad9, a component of the Rad9-Rad1-Hus1 (9-1-1) heterotrimeric complex involved in DNA repair [73]. Phosphorylation of Rad9 at Serine-328 by TLK1B promotes its dissociation from the complex, allowing the cell cycle to resume after DNA damage-induced checkpoint arrest [74] and promoting DNA repair [75]. A proteomic screening has identified 165 binding partners for TLK1B [20]. However, the specific regulation of many of these binding partners in cellular processes is still largely unknown.

Our research group characterized one such proteome target of TLK1 and identified the NIMA-related kinase NEK1 as a significant substrate [76], a critical kinase activating ATR [23,77]. We recently found that ADT in LNCaP cells results in increased expression of TLK1B [78], an essential kinase upstream of NEK1 and ATR, mediating the DDR in androgen-deprived PCa cells [78]. Following DNA damage or replication arrest, we showed that the addition of the TLK1 inhibitor, thioridazine [THD, a phenothiazine (PTH) antipsychotic] [79], impairs ATR and Chk1 activation [20] and foretells the existence of a targetable ADT-TLK1-NEK1-ATR-Chk1 DDR pathway since its abrogation leads to apoptosis [78]. Inhibition of the TLK1-NEK1 axis with THD suppresses the DDR and repair [78,80]. Consistent with our observations, THD was previously shown to improve therapeutic response with DNA-damaging agents [79,81,82,83,84]. What is noteworthy in our work is that the concomitant inhibition of TLK1 along with ADT results in the specific killing of androgen-sensitive (AS) PCa cells instead of the more generalized, and thus more prone to side effects, inhibition of ATR or ATM. The increased expression of TLK1B following the shift of LNCaP cells to CSS-medium, which we deem to be a rapid pro-survival mechanism, leads to the activation of NEK1 and the DDR that can be optimally targeted in PCa. Indeed, treatment with THD bypasses cell cycle arrest elicited by ADT and causes unencumbered replication-induced DNA damage, leading to apoptosis in vitro and in mouse xenograft models [78]. Targeting the TLK1-NEK1 axis is a proposed novel therapy for PCa combined with ADT. An analysis of five independent studies of PCa incidence in individuals with schizophrenia revealed a significant decrease in the Standardized Incidence Ratio ranging from 0.49 to 0.76 (95% CI) [85]. After all possible variables were considered, the most compelling study specifically pointed to the use of PTH as the likely cause [86,87]. We propose this is due to the inhibition of TLKs by PTH antipsychotics.

## 5. TLK1 in PCa Progression

PCa is diagnosed in ~200,000 US men annually, resulting in approximately 30,000 deaths. ADT with anti-androgens is the standard of care for patients with locally advanced prostate cancer, metastatic prostate cancer, and biochemically recurrent disease after failure of localized treatments. ADT is known to provide remission of the disease, as best evidenced by a decline in prostate-specific antigen (PSA) in about 90% of patients. However, after a mean time of 2–3 years, the disease progresses despite continuous hormonal manipulation [3,4]. This type of cancer is known as mCRPC, which has a poor prognosis with a mean survival of only 16–18 months [5] with slight improvement with chemotherapy [88]. The best window of opportunity is before the development of mCRPC [6], and this requires a clear understanding of the process of PCa cells’ mechanisms of adaptation to ADT. The best-characterized model so far for studying this is the LNCaP. Androgen deprivation of LNCaP cells results in loss of AR function with compensatory pro-survival activation of mTOR [89] mediated by loss of the mTOR inhibitor, FK506-binding protein 5 (FKBP5), which is an AR-regulated gene [90], and concomitant implementation of a cell division arrest by activation of the DDR mediated by ATR-Chk1 [91] or ATM-Chk2 [92]. However, what signals the DDR and ATR activation is poorly understood [93]. Then, after a quiescent period of ADT adaptation of 2–3 weeks, androgen-independent colonies begin to form [94]. We have reproduced these effects in TRAMP-C2 cells, another line that recapitulates the conversion from AS to AI growth typically observed in most PCa patients following ADT. An attractive strategy to prevent this process would be to bypass the cell cycle arrest via inhibition of ATM or ATR, causing the cells to undertake replication with damaged DNA that would cause mitotic catastrophe, a strategy that was implemented in LNCaP treated concomitantly with bicalutamide (BIC) and ATM inhibition [92]. Nevertheless, a limitation of this approach is how to make the inhibition of ATM or ATR specific to PCa cells. Figure 2 represents our working model of how TLK1/1B are key DDR regulators that allow PCa cells to survive ADT and reprogram into AI growth. Elucidation of optimal pharmacological targeting of these pathways will lead to more effective and potentially curative therapy for hormone naïve PCa and pre-empt CRPC.

We found that androgen deprivation in LNCaP cells results in a rapid, mTOR-dependent, rapamycin-suppressible, substantial increase in the translation of TLK1B, whereas its mRNA does not change [78]. Similar results were obtained with TRAMP-C2 cells [78] and, more recently, in an AR+/PDX adenocarcinoma model (NSG-TM00298) [95] and also with VCaP cells, which also convert from AS to AI after ADT [96], suggesting that this is likely a general response of AS PCa cells. We previously published an extensive study showing that the translation of TLK1B mRNA is highly dependent on AKT-mTOR-4EBP1 owing to its inhibitory 5’UTR that is not present in TLK1, as confirmed by polysomal distribution [68]. We showed that 4EBP1 becomes rapidly phosphorylated in LNCaP after ADT [78]. Notably, the TLK1 gene was identified by weighted correlation network analysis (WGCNA) as a critical driver of PCa with poor prognosis [97], and this is particularly evident for low Gleason scores (https://ualcan.path.uab.edu/cgi-bin/TCGA-survival1.pl?genenam=TLK1&ctype=PRAD; accessed 12 October 2022). However, since the expression of TLK1B is controlled mainly at the post-transcriptional level, it may have been primarily underestimated in transcriptomics analyses of PCa. TLK1 activates NEK1 by phosphorylating T141, which is adjacent to the activation loop [98], and we established that this mechanism is an early DDR mediator upstream of ATR and Chk1 upon oxidative damage or replication arrest [20], extending earlier studies on NEK1 [23]. We reasoned that ADT activated a similar pathway upon AR inhibition, leading to a pro-survival cell cycle arrest. cBioportal analysis of NEK1 revealed upregulation in 33% of patients and gene amplification in 12% of patients with CRPC-Neuroendocrine PCa (NEPC, a type of AR-negative CRPC). Moreover, NEK1 mRNA is upregulated in 21% of patients with metastatic PCa. Finally, NEK1 is upregulated in PCa patients in the PAN-CANCER analysis of the whole genome dataset [78]. It is tempting to speculate that NEK1 may be involved in some mechanism(s) leading to the survival of PCa cells, particularly at the stage of mCRPC development and significantly in AR-negative NEPC. We have produced a highly specific antiserum to pNek1-T141 and demonstrated that THD (an inhibitor of TLK1) suppresses T141 phosphorylation/activation [78]. We also showed that the extent of pNEK1-T141 correlates with the Gleason Score (GS) in human PCa tissue microarrays (TMA) [95]. Moreover, overexpression of kinase-dead TLK1 suppresses the phosphorylation of NEK1 [20], suggesting that only TLKs mediate the T141 phosphorylation of NEK1. Finally, we showed that ADT of AS cells results in increased expression of TLK1B and, in turn, increased levels of pNEK1-T141. Suppression of NEK1 phosphorylation/activation with THD impaired ATR/Chk1 activation (ADT-TLK1B-NEK1-ATR-Chk1, DDR) and its abrogation with THD led to apoptosis in vitro and [78] in xenografts [78].

Extensive studies also suggest the critical importance of the Hippo/YAP pathway in various solid tumors and in mCRPC [99]. However, the fundamental mechanism of YAP hyperactivation in PCa onset and development remains elusive. Importantly, our preliminary studies discovered that TLK1 has a role in the Hippo pathway (via NEK1-mediated upregulation of YAP) and can directly hasten the conversion toward CRPC [49]. Experimentally, we found that 2-fold overexpression of wt-NEK1 (but not the T141A mutant) hastens the progression of LNCaP cells to AI growth [78]. The protective cell cycle arrest mediated by the TLK1-NEK1 DDR pathway seems insufficient to explain the rapid growth recovery observed in BIC-treated cells when NEK1 is overexpressed and suggests that NEK1 may have additional functions. We suspected it might regulate the Hippo pathway. It was reported that ectopic expression of YAP was sufficient to convert LNCaP cells from AS to AI in vitro [100]. NEK1 was found to phosphorylate TAZ specifically at S309 [101], resulting in increased CTGF expression (one of the TAZ/YAP transcriptional targets). TLK1 may regulate the Hippo pathway through its activity on NEK1. We showed that overexpression of the NEK1-T141A mutant results in reduced levels of YAP, along with evidence of an elevated cleaved product (Cl-YAP) [24,49]. Decreased YAP levels and Cl-YAP are also seen in control LNCaP cells treated with 1 μM THD for 1 h (+/− BIC), with similar effects obtained with J54 (our in-house test, 2nd generation TLK1 inhibitor, described later). In contrast, LNCaP cells that overexpress NEK1-wt show elevated expression of YAP and no evidence of Cl-YAP. We also showed that NEK1 interacts with YAP as coIP enriches it, and THD does not alter their affinity. As previously demonstrated [20], the coIP also brought down TLK1. Inhibition of TLK1 with THD or J54 resulted in dose- and time-dependent degradation of YAP in mouse PCa-NT1 cells, showing conservation of function [24]. All this suggests that TLK1 and NEK1 may be novel regulators of the Hippo pathway, leading to higher YAP levels or phospho-driven nuclear translocation. In contrast, reduced TLK1 activity could impair YAP-driven gene expression via inhibition of the TLK1-NEK1 axis, which seems vital for stabilizing YAP and its accumulation. A recent literature search showed that a known PTH TLK1 inhibitor (Cl-Promazine [79]) increases pYAP-S127 and degradation in aggressive (YAP-driven) BCA cells [102]. In further support of our arguments that NEK1 activity is critical for YAP stabilization, we found that CRISPR-mediated KO of NEK1 in the PCa line Neo-TAg1 (NT1) revealed that YAP expression was concomitantly reduced in all the KO-positive clones. Consistently, the expression of several YAP-target genes (e.g., CTGF, Zeb1, Twist1) that drive EMT and invasiveness of these cells was suppressed in the NEK1-KO clones [24], and there was no apparent compensation from other NEKs, which are not reported to act on YAP [103]. We have also tested by qRT-PCR the expression of several of these genes in LNCaP and C2 cells treated with BIC+J54, but in that case, the main effect was a dramatic 70-fold increase in BAX [49], confirming an effect previously reported in response to PTHs treatment of cancer cells [104]. This suggests that any residual YAP switches to the YAP/P73 transcriptional complex that implements the apoptotic program in response to DNA damage [105]. This is in contrast to the reported effects of ADT alone, where in some models, AR inhibition results in YAP transcriptional activation; in turn, transcription of stemness and EMT-driver genes in a TEAD-driven manner induces sphere formation in vitro [99]. Conversely, THD alone can inhibit cell migration via suppression of EMT-related genes such as Claudin1, E-cadherin, N-cadherin, Twist1, Snail3, Slug, FOXC2, MMP3, and MMP9 in HCC cells, though this was hastily attributed to its activity on dopamine receptors [106].

Our study revealed that the TLK1-NEK1 pathway also significantly supports PCa cells’ survival by maintaining mitochondrial membrane integrity [26]. Specifically, NEK1 phosphorylates VDAC1, a protein that regulates the exchange of molecules and ions between the mitochondria and the cell. In high-grade PCa, VDAC1 is often overexpressed [107]. Our investigation demonstrated that interfering with the TLK1-NEK1 pathway could enhance the sensitivity of PCa cells to low doses of the commonly used chemotherapy drug, doxorubicin. This increased sensitivity was achieved by reducing the phosphorylation of VDAC1 at a specific site, Serine-193, which destabilized it. Upon treating cells that expressed a mutant NEK1-T141A with doxorubicin, we observed a higher number of cells in the sub-G1 phase, decreased oxygen consumption, and elevated cytochrome C release in the cytoplasm. These alterations activated the intrinsic apoptotic pathway, a crucial mechanism responsible for cell death [26].

TLK1 promotes cell survival through various other pathways independent of NEK1. One such pathway involves TLK1’s interaction with AKTIP, a protein that facilitates anti-apoptotic responses. TLK1 phosphorylates AKTIP at specific sites, enhancing its ability to anchor AKT and PDK1 to the plasma membrane, where AKT is subsequently fully activated through phosphorylation by the mTORC2 complex [27]. Once activated, AKT inhibits pro-apoptotic factors such as BAD, caspase 9, and FoxO3, promoting cell survival and contributing to the progression of CRPC. However, the impact of TLK1-mediated AKTIP phosphorylation on CRPC drug resistance requires further investigation.

Additionally, we have discovered a new interaction between TLK1 and MK5, which promotes the motility and invasion of PCa cells [108]. MK5 is a known kinase involved in cell motility, and TLK1 phosphorylates MK5 at three specific sites, enhancing its catalytic function. One of these phosphorylation sites (Serine-354) by TLK1 is present in all significant PCa cell lines, regardless of their dependence on androgens. Furthermore, ADT increases the level of pMK5 S354 in LNCaP cells, and this phosphorylated MK5 protein progressively increases in tumors with higher grades and nodal metastatic scores in both TRAMP-C2 mouse prostate tissue and TMA. These findings suggest that the TLK1-MK5 signaling axis may be associated with increased PCa aggressiveness. Lastly, disrupting the TLK1-MK5 axis through TLK1 or MK5 inhibition significantly reduces the motility of PCa cells, as observed in various PCa cell lines, and reduces colonization of lungs after tail-vein injections [108].

## 6. Phenothiazines (PTH) as TLK1 Inhibitors

Our 2013 research paper highlighted the importance of identifying antipsychotic PTH compounds with similar structures that can inhibit TLKs in laboratory settings [79]. These significant findings originated from a screening procedure involving the examination of three compound libraries, each comprising approximately 6000 compounds [79]. THD exhibited the highest inhibitory activity among the compounds tested. We also reported that THD had some growth-inhibiting effects on various PCa cell lines, either AS or AI [78]. However, it is essential to note that THD is a known dopamine antagonist associated with undesirable side effects. Its use in treating schizophrenia was discontinued due to the increased risk of long-term cardiac arrhythmia and potentially irreversible extrapyramidal toxicity [109]. The adverse effects of THD have been attributed to its anti-dopaminergic activity. Therefore, there was a strong need to design and develop next-generation TLK inhibitors to overcome these side effects and progress toward clinical applications.

Our goal was to create new PTHs that effectively treat PCa while avoiding the harmful side effects of THD. By employing in vitro kinase assays and conducting docking studies, we identified a novel PTH derivative, J54, with selective TLK inhibition and seemingly devoid of toxicity. Through our collaboration with a medicinal chemist and structural modelers, we have synthesized, modeled, and tested several custom-made PTH inhibitors of TLK1 [110,111]. Though the cellular target for the PTH-antipsychotics for cancer treatment remains unknown, studies have suggested their effects by inhibiting dopamine receptors [112,113]. However, our preliminary work suggests that J54 binds weakly to the dopamine receptor (DR2) and has very limited anti-dopaminergic effects in mice and *C. elegans* [110]. The homology model for the TLK1 was created and refined by a 1 µs molecular dynamics (MD) simulation. J54 and THD were docked into the final MD-refined model. In the docked pose of compound J54, good interactions with the hinge region residues of TLK1 were exhibited; the morpholino head forms hydrogen bonds with Asp122, with a distance of 1.68 Å from the Asp122 carboxylate Oδ to the ligand NH group.

A 100 ns MD simulation of this TLK1-J54 complex indicated the pose was stable, further evident from a favorable total binding free energy ∆G_bind_ = −39.7 kcal/mol, computed by the MM/GBSA method [114]. THD was docked in the same TLK1 pocket but did not form hydrogen bonds with the protein; the free energy of binding is 9.9 kcal/mol weaker [110]. We note that this could partly arise from the methylthio group of THD, which prevents it from entering the hinge region to the same extent as J54. In contrast, a comparison of THD, J54, and risperidone revealed that J54 was the weakest for binding to the dopamine receptor (DR2). We tested some 18 other PTH in all and found some inactive as TLK inhibitors, but several others worked well, while scaffold-modified indoles showed variable inhibition at 5 µM. Docking and MD studies revealed that several PTH-ring-N-substituents prevent some compounds from entering the hinge region. We now have more than 30 different PTH derivatives. We continued to study J54 and commissioned a Kinome Scan (DiscoverX), which revealed that only TTK out of 463 kinases (including LATS1/2 and MST1/2) was inhibited in addition to TLK1 and TLK2 [110]. In addition, a competitive assay with radiolabeled dopamine onto recombinant DR2 (human D1 and D3) revealed that J54 is weakly competitive compared to THD: Ki~450 nM for purified receptor protein compared to ~20 nM (carried out by XenoTech-Sekisui, which for THD was consistent with the literature [115]). Such inhibition would be expected to be minimal in vivo at circulating levels. A direct comparison in TRAMP-C2 cells of the inhibitory effect of THD vs. J54 in suppressing pNek1-T141 revealed similar potency (similar results were obtained for LNCaP and VCaP), but the effect of J54 lasted longer (3 days compared to 1 day for THD) which suggests prolonged inhibition of TLKs. Inhibition of IP’d TLK1 from LNCaP cells was evident at 100 nm. J54 also had favorable PKs, where plasma concentrations of 100 ng/mL were reached 2 h after IP injection (5 mg/kg), and ~6 ng/mL was still detected after 24 h [110] (better than THD [116]). While some of our newer compounds are more potent (some in the low nM) and more soluble than J54 [111], we are behind (compared to J54) on their specificity characterization and their in vivo PK/PD and tumor regression potency. Moreover, while the additional targets and possible side-effects of PTHs are generally known (e.g., DR2 and hERG channel), the compounds with indole scaffold modification have less medical literature and records of FDA approval. However, we plan to continue their characterizing, while J54 will be pursued as the lead compound in the proposed pre-clinical studies [J54 is now commercially available (TLK1 inhibitor J54|TLK1 inhibitor|Probechem Biochemicals)].

We tested J54 with the panel of PCa cells in the androgen-containing medium by proliferation assays [110]. The effect was a weak dose-dependent inhibition with maximal efficacy of around 18 µM. Note, however, that RWPE1 (a healthy prostate cell line with similar TLK1 expression [117]) was insensitive to J54, suggesting that J54 may be a more targeted anti-cancer agent or the TLK1-NEK1 axis may remain critical even for the CRPC cells. In contrast to these mild effects, combination treatment of the AS cell lines—LNCaP, VCaP, and TRAMP-C2—with bicalutamide (BIC) and J54 resulted in 4-5-fold suppression in colony formation. Note that VCaP cells were 60% inhibited by J54 alone, consistent with their elevated replication stress-driven DDR and checkpoint activation [118], which could lead to apoptosis when suppressed with J54. Clonogenic assays cannot distinguish the effects of a DNA-damaging agent, which can result in either an impaired or delayed resumption of growth (cell division) or loss of viability due to increased killing of the initial population. We thus measured in LNCaP and TRAMP-C2 the early change in cell number (MTT assay) over 72 h with J54 (only two divisions). The results indicated an actual loss in cell counts with the dose. These results reinforced our central thesis that inhibiting the TLK1-NEK1 axis is the most effective regimen in AS cells when combined with antiandrogens. Indeed, cell cycle analysis of LNCaP and TRAMP-C2 cells treated with BIC, J54, or combination for 24 h showed a substantial increase in the fraction of apoptotic cells only in the combination group. LNCaP overexpressed a dominant-negative KD mutant of TLK1 arrest in G2/M with BIC before dying [110] and failed to form tumors, confirming the critical importance of TLKs for PCa progression. These cells also display markedly reduced pNEK1-T141 and YAP. The TLK1-KD inhibits all TLK isoforms through dominant dimerization, while specific TLK1 knockdown in HeLa (also expressing TLK2) results in complete loss of pNEK1-T141 [24]—onlyTLK1/1B phosphorylates NEK1. We also found that bypass of the DDR imposed by ADT via block of the TLK1-NEK1-ATR-Chk1 axis with THD [78] or J54 [110] results in abrogation of the checkpoint and accumulation of replication-dependent DNA damage followed by induction of apoptosis. TLK1 and NEK1 also cooperate in preventing apoptosis by maintaining mitochondrial membrane potential integrity through the phosphorylation of VDAC1 [26]. Combining BIC and THD [78] or J54 [110] drives remarkable regression of LNCaP tumors via suppression of the pNEK1-pATR-pChk1 DDR pathway [110]. The control group showed progressive exponential growth, and so did the BIC group after a 12-day lag (CRPC progression). In contrast, treatment with J54 alone showed significant suppression of tumor growth and weight; but the combination resulted in substantial regression of tumor volumes and weight. This proposed mechanism appears evolutionarily conserved, as both LNCaP and TRAMP-C2 gave similar results. Additionally, xenografts of LNCaP cells overexpressing a Nek1-T141A mutant were durably suppressed with BIC (did not convert to AI) [78], suggesting that the activity of Nek1 is critical during adaptation to ADT and progression toward CRPC. We have also shown in LNCaP that overexpressing wt-Nek1 counteracts the growth inhibition by BIC and loss of viability with THD in vitro, whereas the Nek1-T141A hypoactive mutant does not.

We also performed a general assessment of mice following injections of J54 up to 30 mg/kg-IP [110]. We noticed no toxicity (even after gross inspection of organs at necropsy), no decrease in body weight, and no behavioral changes (no lethargy or extrapyramidal twitches sometimes observed with THD). To get a better and quantitative assessment of the possible behavioral effects of J54, we have used *C. elegans* which has a simple but complete nervous system and has been well characterized for its responses to antipsychotic drugs, including actions at serotonin or dopamine receptors [119,120]. In three preliminary studies, J54 had much weaker behavioral effects than TFP or risperidone, attributable to DR2 activity. For example, while seeding *C. elegans* on plates containing dopamine results in their immobility (via DR2 signaling), concomitant inclusion of DR2 antagonists like risperidone or TFP suppressed their immobility. In contrast, J54 had a very weak effect in suppressing immobility [110].

## 7. Discussion

Early-stage, organ-confined PCa is generally well controlled with standard therapy, and even more invasive disease can be managed significantly with current ADT modalities. Unfortunately, once cancer has become resistant to these regimens (mCRPC), there is no effective therapy beyond a few additional months. Even though there is a substantial amount of research aimed at understanding the process of conversion to AI growth of PCa cells, there is still limited knowledge of this process, and most advances in the field have been made, regardless, on developing better androgen receptor-targeted therapeutics. Substantial work has been carried out in studying and targeting the DDR during conversion to CRPC [93,121], but targeting the critical components of the DDR (like ATR/ATM and their effector kinases) or DNA Repair pathways (like the use of PARP inhibitors) do not offer optimal targeted specificity to PCa cells. In addition, the mechanism of DDR activation following ADT was unclear. In comparison, we have identified a better target with TLK1/1B and its rapid and specific elevation following the ADT. Thus, novel and effective TLK1 inhibitors can potentially alter the landscape of DDR targeting agents in PCa by incorporating novel PCa-specific gene expression pathways (AR↓⇒TLK1B derepression).

## 8. Factual Considerations and Alternative Modalities

TLKs play vital roles in various essential cellular processes, including chromatin formation, DNA replication, repair, and transcription [63,71,122,123,124]. Studies conducted on mice and cells indicate that TLK1 and TLK2 have similar functions in developing and maintaining the genome’s integrity [125]. However, specific mutations in TLK2 have been observed in individuals with neurological disorders and intellectual disabilities [64,126], possibly due to its crucial involvement in placental development [127]. Moreover, frequent amplification of TLK2 has been found in luminal breast cancers that express the estrogen receptor (ER). In pre-clinical models, inhibiting TLK2, either alone or in combination with tamoxifen, significantly hindered the growth of MCF7 xenograft tumors [128]. These findings highlight the potential therapeutic benefits of targeting TLK2 in treating breast cancer.

Early studies suggested Asf1 as the principal target of TLKs and implicated them in chromatin assembly roles [64]. However, subsequent work demonstrated that TLK knockdowns and Asf1 KD phenotypes differ regarding chromatin and checkpoint activation [129]. More importantly, Jessica Tyler conclusively demonstrated that Asf1-S192 phosphorylation previously attributed to TLK is phosphorylated by DNA-PK in a fashion utterly different from that known for TLK activity after DNA damage [130]. Difficulties in phosphorylating Asf1 in vitro were also acknowledged. Asf1 is essential in embryos, while TLK-KO animals are viable and fertile [125]—one reason why TLKs can be therapeutically targeted. Our recent quantitative analysis of direct interactors failed to detect Asf1 among the top 100 interacting proteins [20], suggesting that other substrates are much more significant [80], although we should emphasize that we used as bait the TLK1B variant that lacks the NTD that was found to be essential for binding Asf1 [131]. Finally, neither THD nor J54 appear to cause any replication defects or chromatin decompaction linked to checkpoint activation—in fact, they attenuate ATR from replication stress [110]. While we have seen that shRNA to TLK1 can cause some problems in some cells attributable to condensation of chromosomes and mitotic defects [70], there are no reports of this for THD (or J54 so far), and there are few reports of intestinal issues or blood count problems except for the excessive doses.

Previously, TLK2 kinase inhibitors were discovered through a screening process involving commercially available small molecules [128,132]. However, the potency of these inhibitors, their effects in vivo, and their potential off-target effects have yet to be documented. Mortuza et al. (2018) demonstrated that TLK2 kinase activity was entirely suppressed by non-specific kinase inhibitors, Staurosporine, and Nocardiopsis [67]. Subsequent experiments evaluated the effectiveness of various inhibitors that target CDK1 (CGP74541A) and GSK3 (Inhibitor XIII), as well as the Indirubin derivatives E804 and indirubin-3’-monoxime, against TLK2. Except for indirubin-3’-monoxime, these compounds significantly inhibited TLK2 kinase activity. Indirubin, a bioactive compound in the traditional Chinese medicine formulation called Danggui Longhui Wan, has demonstrated promising clinical outcomes in patients with chronic myelocytic leukemia [133]. As a result, extensive research has been conducted on indirubin. In a study conducted by Lin et al. (2020), the growth-inhibitory effect of three PTHs [perphenazine (PPH), trifluoperazine (TFP), and Thioridazine (THD)] was examined in a group of nine breast cancer patients and immortalized cell lines [134]. These cell lines exhibited varying levels of TLK2 copy number and protein levels. The researchers observed a significant reduction in cell proliferation at a concentration of 5 μM for PPH and TFP and 2.5 μM for Thioridazine, specifically in high TLK2-expressing cell lines such as MCF-7, MDA-MB-361, and MDA-MB-157. On the other hand, low TLK2-expressing cell lines, including BT474, MDA-MB-231, and BT20, showed minimal response to the PTHs, with no inhibitory effect observed in the three tested immortalized cell lines.

While we aimed to demonstrate that J54 operates by inhibiting the ADT-activated TLK1-NEK1-ATR-Chk1 DDR pathway, we cannot entirely dismiss the possibility that the observed tumor regression effects may result from the inhibition of alternative targets. Another PTH, similar to J54 but lacking DR2 antagonism, was found to exhibit antitumor activity by activating PP2A. J54 might inhibit essential kinases such as ATR, ATM, and mTOR, which were not assessed in the SCANMAX panel. In our experiments, we observed ATM activation in cells treated with BIC+THD or J54, while THD has been reported to indirectly or directly inhibit mTOR [135,136]. The combined activation of PP2A and inhibition of mTOR could synergistically suppress pancreatic adenocarcinoma growth [137]. TTK, another kinase inhibited by J54, is known to play critical roles in mitotic progression and has been implicated in pancreatic cancer [138]. Additionally, it is crucial to consider that J54 might target AR [139], complicating our findings’ interpretation. However, the significant tumor regression observed in mice treated simultaneously with J54 and BIC, a more potent anti-androgen, suggests that the two drugs likely act on distinct pathways.

Together, these findings substantiate the potential use of PTH derivatives as alternative treatments for cancer by pharmacologically targeting the TLK1/2-mediated DDR pathways. Moreover, alongside other studies, our research adds to the comprehensive understanding of the molecular mechanisms of TLKs and positions them as attractive kinase targets for therapeutic interventions against cancer.

## 9. Bicalutamide vs. Enzalutamide: Potential Therapeutic Competition and Comparative Analysis

For more than 50 years, the focus of research and treatment for PCa has revolved around ADT and developing therapies that target AR through various mechanisms [140]. However, these treatments eventually fail as PCa adapts through different pathways [141], leading to the terminal phase of the disease, mCRPC. Referring to the information we discussed in previous paragraphs, we explained how ADT leads to increased production of TLK1B, a crucial mediator in facilitating PCa adaptation to ADT. This translational enhancement of TLK1B is achieved through the compensatory activation of the mTOR-4EBP1 pathway. TLK1B, in turn, promotes cell cycle arrest by activating a kinase cascade involving TLK1-NEK1-ATR-Chk1. This cell cycle arrest prevents PCa cells from entering the cell cycle under unfavorable growth conditions and reprograms them to adapt to AI growth via the NEK1-YAP/AR-CRPC conversion pathway.

Interestingly, one of the most significant findings in initial in vitro studies, later confirmed in an LNCaP xenograft model, was the combination of a specific TLK1 inhibitor called J54 with the first-generation non-steroidal antiandrogen, BIC, or the second-generation antiandrogen Enzalutamide (ENZ). When BIC alone was used, it induced a cell cycle arrest in the G1 phase in PCa cell lines LNCaP or TRAMP-C2. However, when combined with J54, this arrest was bypassed due to impairment of the TLK1-NEK1-ATR-Chk1 axis. Instead, the cells temporarily accumulated in the G2/M phase (the next possible checkpoint after DNA damage and response activation) and then underwent apoptosis (sub-G1 cells) due to a catastrophic attempt at mitosis (Figure 3, Top panel).

Treatment with J54, BIC, or ENZ showed similar induction of apoptosis in androgen-sensitive (AS) cells (Figure 3, Top and Bottom panels). Specifically, there was no difference in apoptotic induction when using BIC (Casodex^®^) compared to ENZ (Xtandi^®^) in combination with J54, at least in vitro. Neither BIC nor ENZ alone induced apoptosis in LNCaP cells; instead, they induced quiescence, as commonly observed [142]. The mechanism of G1 arrest induced by either antiandrogen is primarily attributed to the role of the AR as a replication licensing factor [143]. The bypassing of this arrest with J54 leads to entry into catastrophic mitosis, combined with the challenges of adapting to the lack of AR activity through the Nek1-YAP potentiation of AR integration signals when androgens are scarce or outcompeted by either BIC or ENZ.

**Figure 3 ijms-24-11100-f003:**
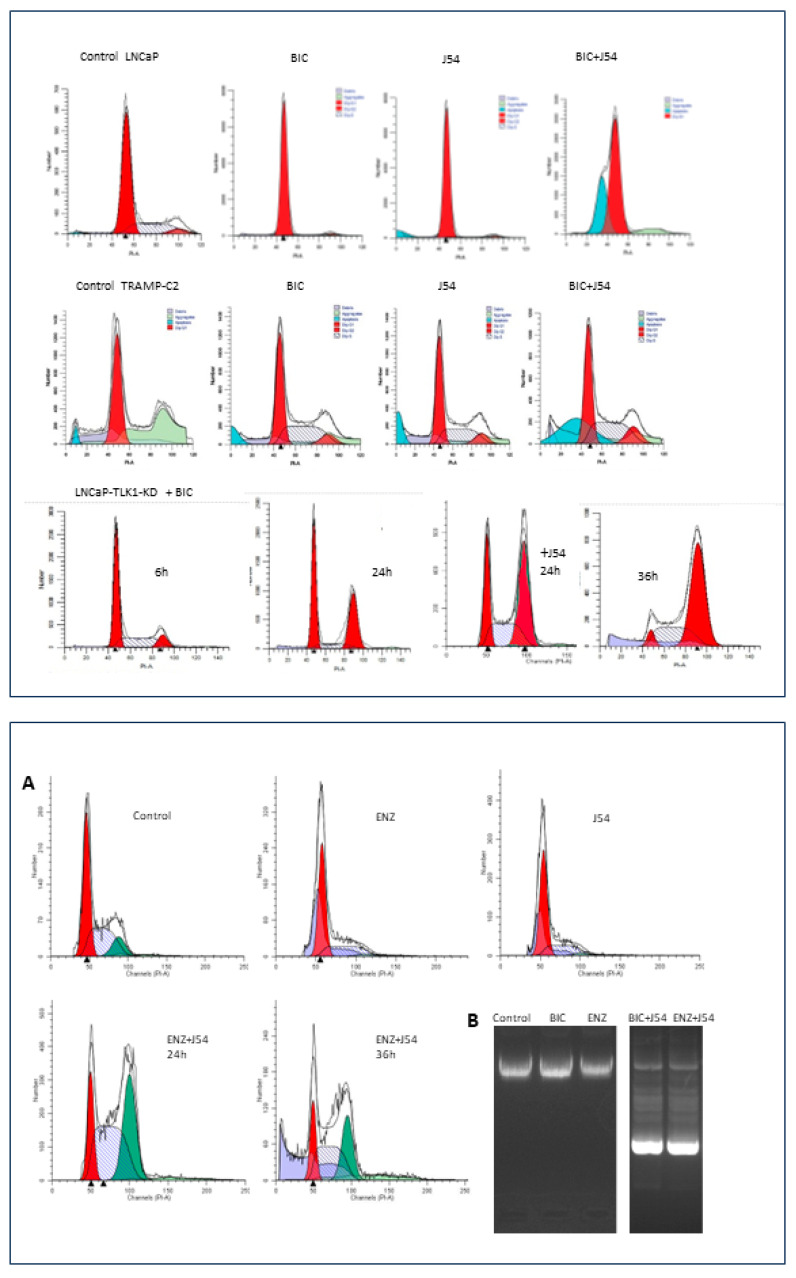
**Top**. Cell cycle analysis by PI-FACS of human LNCaP and mouse TRAMP-C2 cells treated with Bicalutamide (BIC), J54, or combination (5 μM each). Likewise, LNCaP cells expressing a dominant TLK1-Kinase, Dead, when treated with BIC, arrest instead in G2, followed by apoptosis after 36 h [110]. FACS analysis was carried out as described in [20]. **Bottom**. (**A**) Cell cycle analysis by PI-FACS of LNCaP cells treated with Enzalutamide (ENZ), J54, or a combination (5 μM each). (**B**) Apoptotic ladder in ENZ or BIC+J54 for 48 h, which was generated as described in Apoptosis DNA fragmentation analysis protocol|Abcam.

Based on our observations, it must be pointed out that, from an economic perspective, BIC (Casodex^®^) is a more cost-effective option for PCa patients than ENZ (Xtandi^®^) or Apalutamide (Erleada^®^) in the United States. However, when considering the modest improvements in overall survival (OS) and progression-free survival (PFS) observed in ENZ-treated patients compared to those treated with BIC, these benefits seem limited to white patients only [144,145]. Including J54 as an early addition to the treatment regimen in a clinical trial, setting may be beneficial, potentially nullifying the relative benefit and racial differences observed.

## 10. Perspective and Future Outlook

Highlighting the importance of PARP inhibitors (PARPi) in treating PCa with BRCAness characteristics reveals novel insights into PCa biology. Intriguingly, PARPi has demonstrated efficacy in cases with BRCAness [146,147] and, in some cases, without this feature [148,149]. Moreover, it is worth noting that African American (AA) patients derive more significant benefits from PARPi compared to Caucasians (CC). Recent genome-wide association studies (GWAS) have uncovered a significant distinction: while CC patients generally exhibit a lower frequency of mutations in genes associated with the nucleotide excision repair (NER) pathway, around 89% of AA-PCa patients have at least one mutation in these genes. This finding implies that therapies targeting DNA damage, such as radiation and genotoxins, could be particularly advantageous for AA patients in effectively eradicating cancer cells. Consequently, this sheds light on the fact that prostatectomy is often unnecessary for many patients, and even when recommended, it may not yield optimal outcomes.

In response to these insights, numerous researchers, including our team, are actively developing alternative treatment approaches involving combination therapy utilizing cisplatin and a novel DNA repair inhibitor (J54 in our case). Such an approach holds promise for circumventing conventional ADT or using antiandrogens, which often involve chemical castration and are generally undesirable for most men. Additionally, it may offer an alternative to late-stage chemotherapy relying on taxanes. Our research is centered around investigating the role of TLK1 in the DDR process, its anti-apoptotic function mediated through the NEK1-VDAC1 axis and AKT, and its influence on the activity and stabilization of YAP. Significantly, these various mechanisms converge on actionable targets that can be effectively addressed by introducing TLK inhibitors.

## Figures and Tables

**Figure 2 ijms-24-11100-f002:**
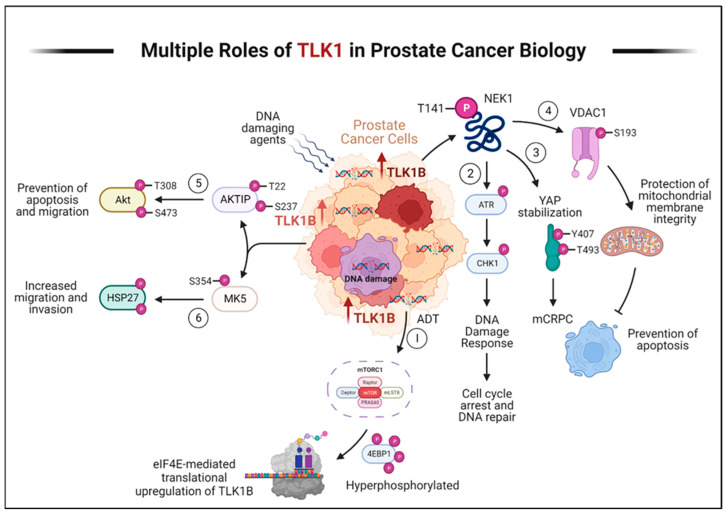
TLK1 plays multiple crucial roles in prostate cancer progression and therapy resistance. Here, we describe the various mechanisms by which TLK1 contributes to these processes. (1) ADT triggers the activation of mTORC1, which activates 4EBP1, causing the release of eIF4E. Excess eIF4E initiates the translation of TLK1B. (2) TLK1/1B, once produced, activates NEK1 by phosphorylating it at T141. Activated NEK1 activates the ATR-Chk1 DDR signaling cascade. The activation of DDR promotes DNA repair, which aids in the resistance to DNA-damaging therapeutic agents. (3) Through TLK1-NEK1 signaling, YAP is phosphorylated on Y407 and T493 residues, stabilizing it. This phosphorylated YAP binds to TEAD or other transcription factors (TF) and relocates to the nucleus, evading proteasomal degradation. The accumulation and stabilization of YAP contribute to the progression of CRPC and resistance to drugs. (4) The TLK1-NEK1 axis also plays a role in phosphorylating VDAC1 at S193, which helps maintain the integrity of the mitochondrial membrane and inhibits intrinsic apoptotic signaling. (5) TLK1 directly phosphorylates AKTIP on T22 and S237 residues, activating AKT. This activation of AKT promotes pro-survival and pro-migratory signaling. (6) TLK1 also interacts with and phosphorylates MK5, enhancing its catalytic activity towards HSP27, a substrate of MK5. This increased activity of MK5 leads to enhanced prostate cancer cell migration, invasion, and metastasis. Created with BioRender.com.

## Data Availability

All data is available upon request and previously published.

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
