# Peer review of "Targeting Prostate Cancer, the ‘Tousled Way’"

_ijms, 2023, doi:10.3390/ijms241311100_

Round 1

Reviewer 1 Report

The authors have presented a well-written, comprehensive, but also focused review on the mechanisms of targeting androgen-related prostate cancer signaling. Their focus was on the tousled-like kinase 1 (TLK1) regulation of prostate cancer cell adaptation to androgen deprivation therapy (ADT). I like the organization of the different sections of this review, the separation of each section made it easier to follow what the authors were conveying and how the signaling pathways interact to regulate therapy responsiveness.

Some minor suggestions to authors include:
1. Maybe consider using androgen receptor signaling inhibitors (ARSI) instead of androgen deprivation therapy (ADT) to confer with current research on the topic. Several reports have indicated that ARSI is abiraterone, apalutamide, enzalutamide and darolutamide, which the authors discuss in the review manuscript. Sperger et al (PMID: 34197212) and Rajwa et al (PMID: 35907662) indicated that these anti-androgen agents are classified or should be considered as ARSI.

2. Please reference to each of the figures more often throughout the sections. The figures corresponding to the text is not clearly conveyed. Figure 3 is especially not referenced in the text and needs to be because there is actual data sets displayed. The gravity of the data in Figure 3 is lost because it is placed at the end of the document and not discussed in detail for the reader.

3. Figure 2 is a little to complicated, maybe it can be separated into two figures so the concepts are clearly understood.

 Minor editing of English language required

Author Response

  1. Maybe consider using androgen receptor signaling inhibitors (ARSI) instead of androgen deprivation therapy (ADT) to confer with current research on the topic.
    We agree and have included the suggestion in the text.

2. Please reference to each of the figures more often throughout the sections. The figures corresponding to the text is not clearly conveyed. Figure 3 is especially not referenced in the text and needs to be because there is actual data sets displayed. 

We also did notice this omission, and have already taken care of it in the first resubmission.

3. Figure 2 is a little to complicated, maybe it can be separated into two figures so the concepts are clearly understood. 

We considered this suggestion, and agree that the figure is complicated as the disease is, but hav concluded that separating it into 2 figures would possibly makes it even more complicated to follow.

Reviewer 2 Report

Although androgen deprivation therapy (ADT) is an effective treatment for prostate cancer (PC), it can increase androgen-insensitive (AI) PC with a poor prognosis. The authors, through their research, have been able to identify tousled-like kinase (TLK)1 as a crucial mediator in PC becoming ADT-resistant. They discussed, in this review, TLK1 biology, the process of PC transforming into AI disease by TLK1 and the therapeutic targeting of TLK1 to prevent PC progression to a metastatic castration-resistant state. Since treatment options for ADT-resistant PC are limited, the focus of this article is topical. Further, this article is clear and informative. However, addressing the following minor point will improve the quality of the manuscript.

Figure 2 should be placed below the paragraph where it is mentioned the first time (Currently Figure 2 is mentioned on page 2 but the figure is placed on page 6).

There are a few typos here and there, which need to be corrected. For example, the Androgen receptor (AR) is mentioned in short form (AR) on line 3 but in full form on line 6 in the abstract.

Author Response

The only suggestion was to arrange the figures within the text. This is no done. many thanks for the lodative review.

Reviewer 3 Report

This review article summarizes the most recent findings about the role of tousled-like kinase 1 (TLK1) in PCa. The paper is well-structured, carefully and well-written. The authors have clearly focused on the information published over the last years, which has indeed resulted in a timely review. Given the complexity involved, the author has produced many positive and welcome outcomes. The literature review offers a useful overview of current research and policy, and the resulting bibliography provides a very useful resource for current practitioners. Overall, this research is well written, and the content of this manuscript is of major interest and deserves to be published on the pages of IJMS. 

Author Response

There were no suggestions for improvements.  many thanks for the laudative review.